

# Representation of atmosphere induced heterogeneity in land – atmosphere interactions in E3SM-MMFv2

Jungmin Lee[1], Walter M. Hannah[1], and David C. Bader[1]

[1]Lawrence Livermore National Laboratory, 94550, United States

**Correspondence:** Jungmin Lee (lee1046@llnl.gov)

**Abstract.** In E3SM-MMF, where parameterizations of convection and turbulence are replaced by 2-D CRM, there are multiple options to represent land-atmosphere interactions. Here, we propose 3 different coupling strategies: 1) coupling of a single land surface model to the global grid (MMF), 2) coupling a single land instance directly to the embedded CRM (SFLX2CRM), and 3) coupling multiple land instances to each column of the CRM grid (MAML). In MAML (Multi-Atmosphere Multi-Land) framework, a land model is coupled to CRM at CRM grid scale by coupling an individual copy of a land model to each CRM grid. Therefore, we can represent intra-CRM heterogeneity in the land-atmosphere interaction processes. 5-year global simulations are run using these 3 coupling strategies and we find that there are little to no difference whether a land model is coupled to CRM or a global atmosphere. In MAML, spatial heterogeneity within CRM induces stronger turbulence, which leads to the changes in soil moisture, surface heat fluxes and precipitation. However, the differences of MAML from the other two cases are rather weak, suggesting that the impact of using MAML does not justify the increase in cost.

## 1 Introduction

The careful representation of land-atmosphere interaction processes is important to improve the prediction skills of surface weather and climate in numerical models (Betts, 2004b). The key role that land-atmosphere interactions plays in the development of clouds and precipitation is demonstrated in diurnal time scale (Findell and Eltahir, 2003b, a; Gentine et al., 2013; Vilà-Guerau de Arellano et al., 2014; Guillod et al., 2015) and daily to seasonal time scales (Koster, 2010; Hirsch et al., 2014; Dirmeyer and Halder, 2016; Betts et al., 2017). There is also evidence that land-atmosphere interactions can influence the persistence of extreme drought and heatwaves (Roundy et al., 2013; Miralles et al., 2014; PaiMazumder and Done, 2016; Wang et al., 2015; Roundy and Santanello, 2017; Dirmeyer et al., 2021).

However, the complexity of land-atmosphere interactions remains a challenge to weather and climate model development. A contributing factor is that the land-atmosphere interaction processes, which strongly controls the surface water and energy budget, encompasses a multitude of temporal and spatial scales primarily due to the heterogeneous nature of land surface characteristics (i.e., land cover types, soil types, and terrain). There have been several observational studies to better understand the linkage between the land-atmosphere coupling and its influence on the cloud formation and precipitation processes. However, those study results suggest that the land-atmosphere interaction processes are strongly location dependent and difficult to generalize (Betts et al., 1996; Betts, 2000, 2004a; Ek and Holtslag, 2004; Guo, 2006; Jimenez et al., 2014; Teuling, 2017).



Previously many numerical studies used LES or CRM with an interactive land-surface to better understand the land-atmosphere coupling and its influence on the diurnal cycle of clouds and precipitation (Huang and Margulis, 2009, 2013; Rieck et al., 2014, 2015; Rochetin et al., 2017; Lee et al., 2019). Cloud resolving scales are more appropriate to resolve the processes that are important to the cloud formation. However, running the global cloud resolving model is still computationally too expensive to assess the influence of land-atmosphere interaction processes across a various temporal and spatial scales.

The Multi-scale Modeling Framework (MMF) approach that is implemented within a global climate model (GCM) can be a good candidate to assess the impact of land-atmosphere coupling on the clouds and precipitation processes. An MMF embeds a fine-scale cloud resolving model in each cell of the host model to replace the traditional parameterizations for cloud and turbulence. Therefore, the GCM can explicitly represent convective circulations at a reasonable computational cost. At a resolution on the order of 1km or less, the MMF model can explicitly resolve the key processes for the formation of convective clouds without having to run a global cloud resolving model (Grabowski, 2004; Khairoutdinov and Randall, 2003; Khairoutdinov et al., 2005).

Traditionally, in the MMF, the land and atmosphere coupling is implemented between the GCM atmosphere and the land model. The CRM embedded inside each GCM grid does not interact directly with the land surface below (Baker et al., 2019). Instead, the GCM interacts directly with the land surface, and these effects are then felt indirectly by the CRM through the tendencies provided by the host GCM. This strategy does not seem to be appropriate especially when the land-atmosphere coupling plays a key role in the PBL evolution and convection developments, which are explicitly resolved in the CRM. Therefore, in this study, we propose two different ways to model the coupling directly between CRM and the land model for the MMF configuration of DOE's Energy Exascale Earth System Model (E3SMv1; Golaz et al., 2019; Rasch et al., 2019). These two methods differ only by whether the spatial heterogeneity in the land-atmosphere interaction processes is allowed or not. Our approach is based on the single column model study of representing the heterogeneous land-atmosphere coupling by Baker et al. (2019).

The questions we would like to address in this study are: 1) How does the global climatology changes when the land-atmosphere coupling method changes?, and 2) Are these changes in global climate related to the heterogeneity in land-atmosphere coupling?

The remainder of this paper is organized as follow: section 2 describes the model and the 3 land-atmosphere coupling strategies in detail. Section 3 documents the simulation results of how different coupling methods influences the cloud formation and land surface evolution. The summary is presented in the last section.

## 2   Method

### 2.1   Model and Experiment Set up

The model used in this study is DOE's Energy Exascale Earth System Model (E3SMv1; Golaz et al., 2019; Rasch et al., 2019) in the multiscale modeling framework (MMF) configuration. In the MMF approach, a cloud resolving convective parameterization (i.e., super-parameterization) is integrated into a global atmosphere model. In E3SM-MMFv1, parameterizations for clouds and





turbulence in the EAM is replaced by a 2-D CRM that is based on the System for Atmospheric Modeling (SAM; Khairoutdinov
and Randall, 2003) CRM. The description and performance of the E3SM-MMF is documented in Hannah et al. (2020). In this
study, our focus is to explore various strategies to model the land-atmosphere coupling in the E3SM-MMF and analyze their
impact on cloud formation.

We performed three 5-year simulations with E3SM-MMF. The simulations share the same model configuration except for
the land-atmosphere coupling method. Horizontal model resolution of the EAM (E3SM atmosphere model) is about 1.5 degree,
and the number of vertical model level is 72 while the model top extends to 60km. For 2D CRM, we use 32 columns with
a horizontal grid spacing of 2 km. CRM time step is 5 seconds, and the time steps for the EAM physics and ELM (E3SM
land model) are 20 minutes. SST and aerosol concentrations are prescribed using the climatology across the 10-year period
centered year 2000. 1-moment microphysics scheme is used inside the CRM to compute clouds and precipitation processes.
Smagorinsky scheme is responsible for parameterizing sub-grid-scale turbulence in the CRM. Since we use a 2-D CRM, the
meridional component of the wind can be too strong, therefore, misleading the land surface processes if CRM wind fields are
coupled to the ELM. To avoid this caveat, we use the wind velocities from EAM in all three methods. There is also a timestep
difference between ELM and CRM as CRM subcycles with a much shorter time scale (dt) within a single timestep of the EAM
(dT). EAM and ELM share the same time step. Therefore, the CRM state that are passed to ELM are temporally averaged
over dT at the end of the subcycling, and the land surface states do not change while the CRM subcycles. The land model
initial conditions are spun up using 20 years of NCEP reanalysis data. For the radiation scheme, we used RRTMGP (Pincus
et al., 2019). To decrease the computational cost, we used a method where a certain number of CRM columns are grouped
together for the radiation calculations. In our study, we grouped 2 neighboring CRM columns to compute radiation instead of
computing radiation in each CRM grid.

## 2.2 The land-atmosphere coupling strategies for the E3SM-MMF

The coupling between the land and atmosphere models is implemented as the exchange of near-surface atmospheric states
and land surface energy fluxes. The near surface meteorological conditions include downwelling radiative fluxes, temperature,
moisture, wind speed, and precipitation rate from the lowest model level. The land surface fields that are used as atmospheric
lower boundary conditions include surface latent and sensible heat fluxes, and land surface temperature. Also, upwelling short-
and long-wave radiative fluxes are returned from the land model to the atmosphere's radiation scheme. In this study, we explore
three strategies to implement the coupled exchange of water and energy between an atmosphere and a land surface in the E3SM-
MMF, where the complexity increases as numerical simulations of an atmosphere are performed both in the EAM and CRM
components.

The default method to couple the land and atmosphere models is to exchange fluxes directly between EAM and ELM and use
the modified EAM state to force the embedded CRM as shown in Fig.1(a). Therefore, the CRM experiences the effect of land
surface energy fluxes indirectly through the large-scale forcing given by the EAM. This method allows for both atmosphere
and land physics to be updated at the same time scale. E3SM and original version of the E3SM-MMF adopt this strategy. This
method is labeled as 'MMF' throughout this study.

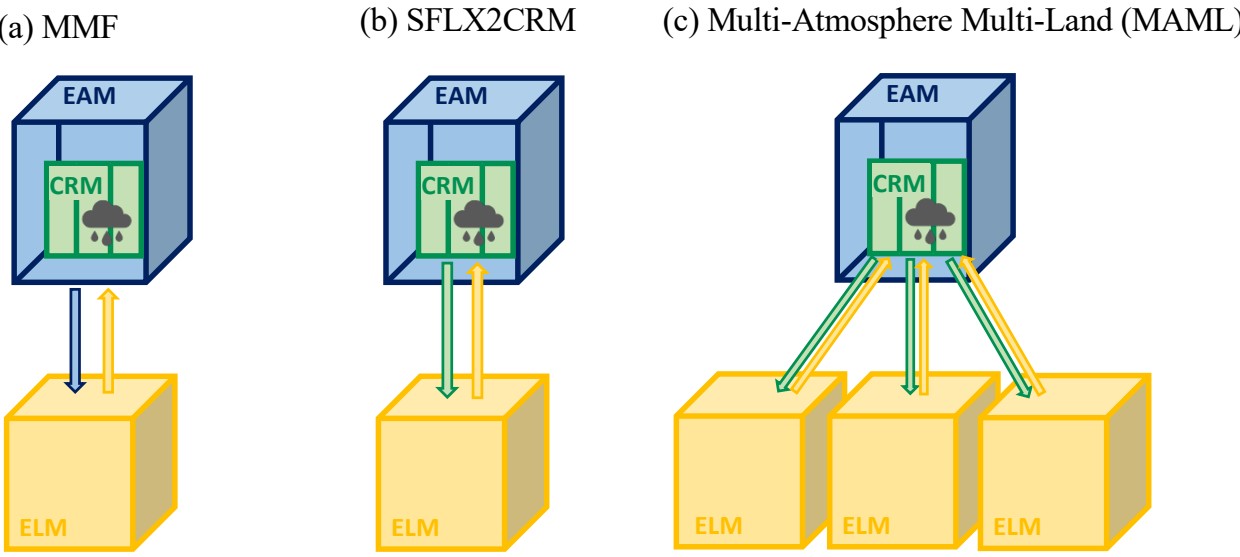

**Figure 1.** Illustrations of three methods to implement the land-atmosphere coupling in the E3SM-MMF. The blue, green, and yellow boxes represent EAM (E3SM global atmosphere model), 2-dimensional CRM atmosphere, and ELM (E3SM land model) grid boxes, respectively. The start and end points of each arrow together with their colors reflect the interface that is created for the exchange of near-surface meteorological conditions and surface heat fluxes between land surface and lower atmospheric boundary layer. In (a), the coupling interface is placed between EAM and ELM, while in (b) the interface is put between CRM and ELM. In (c) each CRM grid is directly coupled to an independent copy of the land grid.

Another method is presented in Fig. 1(b) where a set of near surface meteorological conditions and a set of land surface energy fluxes are exchanged between the CRM and the ELM. The near-surface meteorological conditions are averaged across the
CRM domain, and these spatially averaged fields serve as an input to the ELM. The surface heat fluxes computed by the ELM are applied homogeneously as a bottom boundary condition across the CRM domain. This method allows the development of surface buoyancy driven turbulences in the PBL. This method is called 'SFLX2CRM' hereafter.

The third method is based off SFLX2CRM and allows spatial heterogeneity across the CRM domain by associating each CRM column with a separate instance of ELM. This approach is made possible by the multi-instance functionality of the
E3SM, which was originally developed to perform ensemble simulations. For instance, when the CRM with nx horizontal grids is used, we set up the model to run nx instances of ELM. The coupling is done at the interface between the CRM and the ELM where each CRM grid is coupled to an independent copy of ELM but on the timescale of the EAM time step. This is where the name Multi-Atmosphere Multi-Land (MAML) comes from. These nx copies of ELM are prescribed with the same land surface characteristics. Therefore, this method does not fully represent a significant level of surface heterogeneity.





SFLXCRM and MAML were previously introduced in Baker et al. (2019) for different atmosphere and land models. Baker et al. (2019) ran single column model simulations of the MMF model with changes in how models couple to land surface for Brazilian Forest, while our study runs global simulations to assess the influence of heterogeneous land-atmosphere interaction processes on the global climate. Lin et al. (2023) also presented a method for surface-atmosphere coupling at cloud-resolving scale within the MMF configuration of E3SMv1. Lin et al. (2023) uses the terminology of MAML for the framework where

the land states are averaged across the land instances before coupled to the CRM columns. It is important to acknowledge that Lin's MAML is different from our MAML method.

The cases in this study are referred to as MMF, SFLX2CRM, or MAML following the strategies introduced in Fig.1. The influence of land-atmosphere coupling on the simulated climate is analyzed over land only for the entire 5-year simulation periods.

**3   Results**

**3.1   Climatology Overview**

Figure 2 a,b shows annual mean of daytime surface sensible (SHFLX; left column) and latent heat fluxes (LHFLX; right column) of the MMF simulation. Surface sensible heat fluxes over land experience a strong diurnal cycle, which reaches maximum around local noon. Therefore, only daytime, which is from 6 to 18 local hour, values were averaged to emphasize

the differences between simulations. In comparison to the 5-year climatological mean, only the magnitudes of fluxes increase in the daytime mean, and spatial patterns of fluxes remain the same (not shown).

Spatial distributions of surface sensible and latent heat fluxes show a strong dependency upon the land cover types. Figure 3 is a global map of the prescribed plant functional types (PFTs) that are dominant for each land unit in ELM. The dominant PFT is determined by any PFT of which the coverage for each land unit is greater than 50 %. The areas where there is no

dominant PFT, such as boundaries of vegetation type changes or highly heterogeneous areas, are marked white. It is notable that evaporation in fig.2b is the strongest in the tropical and sub-tropical regions where the primary vegetation types are broadleaf evergreen tropical (PFT=5), broadleaf deciduous tropical (7), cool C3 grass (14), warm C4 grass (15), and crop (16). On the other hand, regions with no vegetation cover (1) produce stronger sensible heating as shown in fig.2a.

Figure 4 a,b show 5-year means of soil moisture (SOILWATER_10CM) and temperature (TSOI_10CM) of the MMF simu-

lation. In general, the MMF case shows that areas with high soil saturation level and low soil temperature, and high evaporative fraction (EF = LHF/(SHF+LHF)), such as Amazon basin, Eastern United States, Tropical Africa, and the Maritime continents, overlaps with the regions with high surface evaporation. Whereas areas with arid climate, such as Australia and deserts in Africa, commonly show low soil saturation level, high soil temperature, and low evaporative fraction. As shown in fig.3, among the vegetated region, areas covered with shorter and less dense vegetation types, such as grass and crop, exhibit higher

soil temperature in comparison to the areas with dense forest. Figure 5 a, b show annual means of near-surface temperature (TSA) and specific humidity (Q2M) from the MMF. The response of surface heat fluxes to the near surface atmospheric temperature and moisture quantity presents a few important features. Similar with the relationship between soil moisture and







**Figure 2.** 5-year averages of daytime (6-18 local hour) mean of (left) sensible heat flux (SHFLX) and (right) latent heat flux (LHFLX) over land only. Top row is from MMF case and remaining two rows are computed by subtracting MMF from SFLX2CRM and MAML, respectively.

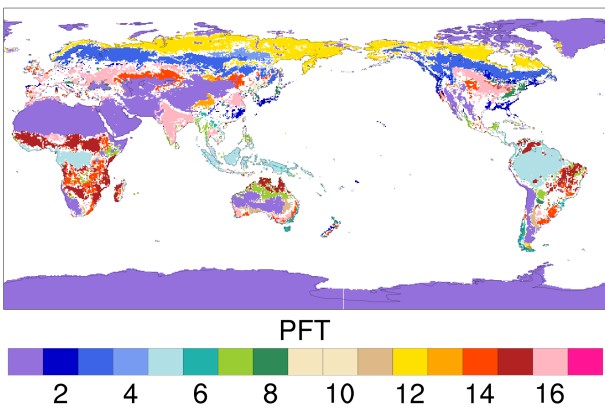

PFT

**Figure 3.** Global map of plant functional types (PFTs) that cover more than 50 % of each land unit grid cell in ELM. The vegetation types that each PFT number indicates are 1- baresoil, 2 - needle-leaf evergreen temperate, 3 - needle-leaf evergreen boreal, 4 - needle-leaf deciduous boreal, 5 - broad-leaf evergreen tropical, 6 - broad-leaf evergreen temperature, 7 - broad-leaf deciduous tropical, 8 - broad-leaf deciduous temperate, 9 - broad-leaf deciduous boreal, 10 - broad-leaf evergreen temperate shrub, 11 – broad-leaf deciduous temperate shrub, 12 – broad-leaf deciduous boreal shrub, 13 – arctic C3 grass, 14 – cool C3 grass, 15- warm C4 grass, 16- crop.

temperature to the surface heat fluxes, the near surface atmosphere also shows dependency on the evaporative fraction. The regions where the evaporative fraction is low, that is where the sensible heat flux is higher than the evaporation, has high at-

mospheric temperature and low moisture content. Therefore, RH is low over areas such as Northern and Southern Africa and Australia, which also overlaps with descending Hadley circulation. On the other hand, the tropical rain forest regions exhibit higher relative humidity in response to the higher latent heat flux. Figure 6a shows that the climatological precipitation over land tends to favor the areas with high atmospheric humidity. Even in Amazon basin, where both terrain effect and local-scale land-atmosphere interaction processes are signification, we see that the location of the enhanced precipitation grossly follows

the area of high evaporation and high soil moisture level. Figure 6b shows the net cloud radiative effect at surface (CRES), that is determined by summing the differences in downwelling longwave radiation between clear-sky and cloudy-sky and in downwelling shortwave radiation between clear-sky and cloudy sky. Surface net cloud radiative effect that has cooling effect indicate the presence of liquid clouds as they reflect more solar radiation than absorbing terrestrial IR. The spatial distribution of clouds overlaps well with the location of precipitation over land, which also has a strong dependency on the location of wet

soil. Therefore, our MMF case shows that regions of high precipitation overlaps the areas with wet soil, especially in a tropical belt.





**Figure 4.** 5-year annual climatologies of (left) top-10cm soil moisture (SOILWATER_10CM) and (right) top 10-cm soil temperature (TSOI_10CM). Top row is from MMF case and remaining two rows are computed by subtracting MMF from SFLX2CRM and MAML, respectively.







**Figure 5.** 5-year annual climatologies of (left) 2-meter surface air temperature (TSA) and (right) 2-meter specific humidity (Q2M). Top row is from MMF case and middle two rows are the changes in SFLX2CRM and MAML from the MMF simulation.



**Figure 6.** Same as Figure 5 but for (left) surface rainfall rate (PRECT) and (right) net cloud radiative effect at surface (CRES).





**Table 1.** 5-year global means over land for MMF, SFLX2CRM, and MAML cases. SHFLX is sensible heat flux, LHFLX is latent heat flux. SOILWATER_10CM is top-10cm integrated soil moisture, SOILT_10CM is soil temperature for top 10cm depth. FCEV is direct evpoaration from vegetation moisture. FCTR is vegetation transpiration. FGEV is soil evaporation.

|  | MMF | SFLX2CRM | MMF |
|---|---|---|---|
| SHFLX ($W/m^2$) | 24.54 | 23.73 | 23.81 |
| LHFLX ($W/m^2$) | 23.18 | 23.19 | 22.51 |
| FCEV ($W/m^2$) | 2.04 | 2.04 | 1.54 |
| FCTR ($W/m^2$) | 11.49 | 11.59 | 11.89 |
| FGEV ($W/m^2$) | 9.64 | 9.56 | 9.08 |
| SOILWATER_10CM ($kg/m^2$) | 27.61 | 27.51 | 27.52 |
| SOILT_10CM ($K$) | 269.44 | 269.41 | 269.63 |
| PRECT ($mm/d$) | 1.09 | 1.10 | 1.09 |

### 3.2 Influences of heterogeneous land-atmosphere interaction in global climate

Global means that are computed over land only suggest that the differences between each case are small (Table 1). The response of surface energy and water cycle to the land-atmosphere coupling in E3SM-MMF is shown in terms of the changes in SFLX2CRM and MAML from the MMF simulation. As shown in Fig.1 through 6 (except for Fig.3), when the land model is directly coupled with CRM, we see many small differences, but no systematic change that is consistent across different regions. Notable differences tend to be collocated with the areas of high net surface radiation, which is approximately equal to the sum of sensible and latent heat fluxes (i.e., tropical-subtropical band).

For a given net surface radiation at each grid cells, the increase in the surface evaporation leads to the reduction in sensible heat flux (Fig.3) While changes in SFLX2CRM are small, MAML shows appreciable amount of reduction in latent heat flux (therefore, increase in sensible heat flux) over the tropical rain forest regions. In comparison to the MMF, a global mean of land evaporation in SFLX2CRM changes by 0.08 % (0.02 $W/m^2$) while the change is -2.9 % (-0.6 $W/m^2$) in MAML (Table 1). The total evaporation from the land surface is composed of 1) direct evaporation from the water present on vegetation surface, 2) transpiration, and 3) evaporation from soil top. Fig. 7 shows ratios of each evaporation component to the total evaporation for each grid cell from the 5-year annual mean. Left column is the relationship for the MMF case, while the other two columns are from SFLX2CRM and MAML cases, respectively. For each row, the relationship is shown for different PFT types. Each row represents the relationship for the broadleaf evergreen tropical type (PTF=5), cool C3 grass (14), warm C4 grass (15) and crop (16), respectively. These 4 PFT types cover the largest area globally, and mostly found in the tropics-subtropics belt.

In Fig. 7, it stands out that the vegetation transpiration makes up most of the total evaporation into the atmosphere regardless of vegetation types. The ratio of transpiration to the total evaporation is the highest for broadleaf evergreen tropical type. Direct evaporation from vegetation in SFLX2CRM show insignificant change relative to the MMF case, while the same field in MAML case decrease overall by 33 % (3.8 $W/m^2$). The source of moisture present on vegetation surface is dependent on

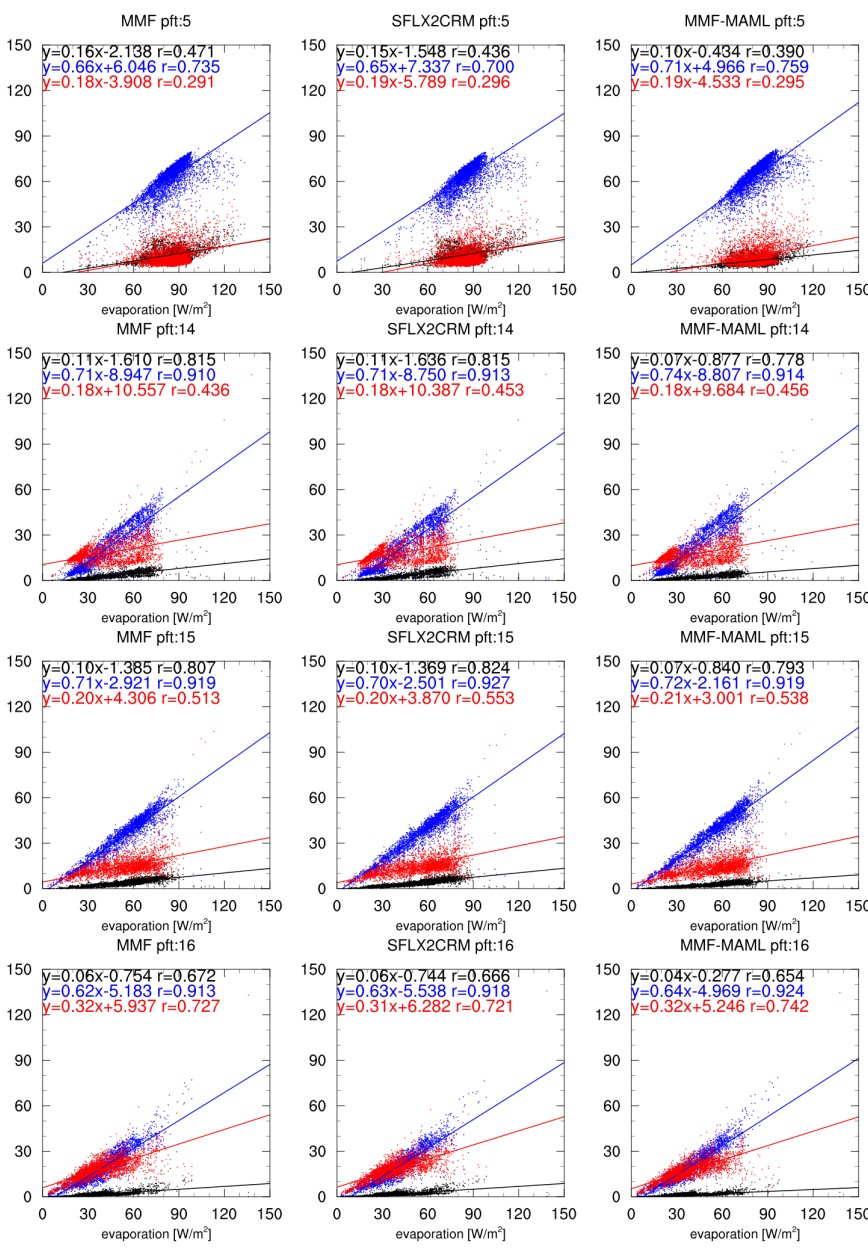

**Figure 7.** Scatter plots of total evaporation (xaxis) and its component (yaxis) - (black)direct evaporation rate from the moisture intercepted by vegetation, (blue) evaporation rate from vegetation transpiration, and (red) evaporation rate from the soil surface against total evaporation. Each column represent different cases: (left) MMF, (center) SFLX2CRM, and (right) MAML. Each row denotes a different PFT type. The equations denote slope and y-intercept of a given linear regression line.





rainfall and nighttime dew formation. The fact that MAML shows reduction in the direct evaporation from vegetation could be related with the decrease in rainfall and increase in temperature, which could prevent dew formation. In addition, MAML

indicates an increase of transpiration approximately 1.31 $W/m^2$ in the tropical regions in Africa and South America despite a slight reduction in the soil moisture in comparison to the other two cases. This feature also suggests that land-atmosphere coupling via MAML method increases surface temperature, therefore, the vapor deficit at the vegetation layer level worsens, which leads to the higher potential for transpiration. Since transpiration of rainforest withdraws soil moisture in the root zone, there was less impact by near surface soil moisture. For tropical rain forest regions, the reduction of latent heat flux in MAML

is due to the reduction of direct evaporation of moisture stored on vegetation surfaces.

In comparison to tropical evergreen broad-leaf trees, C3, C4 grasses and crop are shorter and have lower LAI. These phenology makes grasses and crop types are more sensitive to the soil moisture when determining the total evaporation into the atmosphere. In fact, the ratio of soil evaporation to the total increases for grasses and crop types. The most dominant land type in India is crop-field as shown in Fig.3. MAML has increased soil moisture over India, which results in the increased transpiration

and soil evaporation. Similarly, C3 and C4 grass covered area also shows strong dependency on the soil moisture. For instance, the north side of Congo experiences reduced soil moisture therefore decrease in surface evaporation. While the south side of Congo exhibits increased soil moisture therefore increase in surface evaporation. Fig. 5 shows how the lower-atmosphere condition is influenced by different coupling methods. Comparison with Fig. 6 confirms that precipitation is favorable over the region where the PBL is humid. As suggested before, coupling CRM and land model affects the PBL thermodynamics,

therefore, affects cloud processes that are triggered by PBL turbulence. However, MAML case demonstrates land-atmosphere interactions are at CRM grid scales, and spatial heterogeneity within each CRM domain contributes to warmer and dryer PBL, therefore fewer liquid clouds with less precipitation over land

### 3.3 Spatial Heterogeneity within CRM Column

States of land and atmosphere experience stronger adjustments when CRM is coupled with land, and these adjustments are

enhanced when the spatial heterogeneity of the coupling inside the CRM is allowed in MAML. Figure 9 shows the normalized standard deviation of surface heat fluxes, soil temperature, soil moisture content, 2-meter atmospheric humidity and precipitation across 32 land model instances. We use the standard deviation as a measure of CRM scale spatial heterogeneity. Strong spatial heterogeneity in land surface processes is found where the MAML differs from the MMF the most. Therefore, where the difference between MMF and MAML are the largest in the land surface states and precipitation roughly overlap with the

areas with strong standard deviation. Therefore, the stronger global mean deviation in the MAML can be attributable to the spatial heterogeneity in land-atmosphere interactions.

However, the spatial heterogeneity in lower atmosphere in terms of temperature and moisture seems to be insignificant in comparison to that in the land states. This is due to the mixing inside the CRM homogenizes the atmosphere, while such horizontal mixing processes are missing within 32 ELM instances. This could explain the little changes induced by MAML

land-atmosphere coupling method.





**Figure 8.** Standard deviation of 5-year climatology of (a) sensible heat flux, (b) latent heat flux, (c) top 10-cm soil moisture, and (d) top 10-cm soil temperature , (e) lower-atmosphere relative humidity, and (f) surface precipitation across 32 land instances.



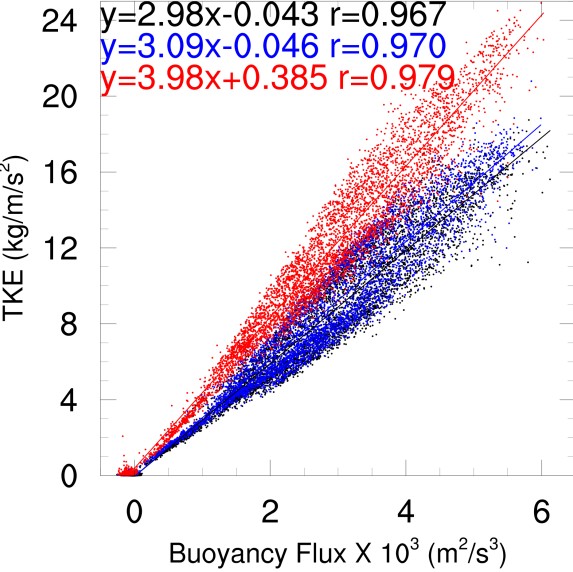

**Figure 9.** Scatter plot of surface buoyancy flux and lowest model level TKE with fitted linear regression line for each case. Each dot represents an annual mean of daytime (6-18 local hour) mean of a single model grid over land. Each case is drawn in a different color. MMF, SFLX2CRM, and MAML are marked in black, blue, and red dots, respectively. Linear regression equation and the correlation coefficient R are written in a color matching that of each case.

### 3.4 Stronger PBL turbulence in MAML

Figure 9 shows the scatter plot between surface buoyancy flux in x axis and production of TKE in the y axis. Each dot represents a 5-year mean of daytime (6-18 local hour) mean values for a given land grid point. Figure 9 shows that there is a linearly positive relationship between surface buoyancy flux and TKE. Therefore, it is shown that stronger buoyancy flux at the

surface results in stronger TKE in all cases. All land points show stronger TKE in MAML for given buoyancy flux, while there is little difference between MMF and SFLX2CRM. This is due to that in SFLX2CRM the ELM computes the buoyancy flux based on the homogeneous meteorological condition within a CRM, which is similar to that of the MMF. On the other hand, in MAML, the heterogeneity in the surface buoyancy fluxes contributes to the stronger PBL turbulence.

Figure 10 presents a globally averaged profile of TKE and liquid cloud water content over land. In MMF, excessive conden-

sation in the lowest model level over land is a globally common feature as shown in Fig.10. However, in the model with CRM with an interactive land surface (SFLX2CRM and MAML) shows a significant reduction in such fog formation in the lowest model level. This is due to the PBL turbulence triggered by the surface buoyancy is effectively transporting the near surface air mass vertically. This process, unlike SFLX2CRM and MAML where CRM is coupled to ELM, was missing when EAM had an interactive land surface (MMF) as these turbulent transport processes in response to the land surface heating were not





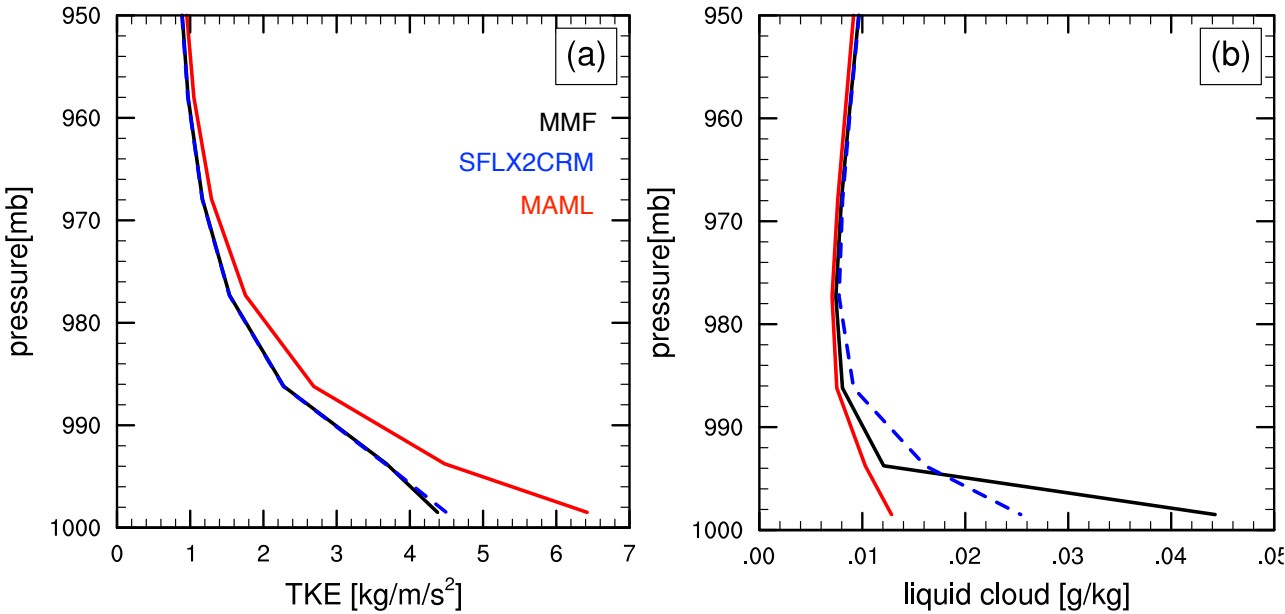

**Figure 10.** Globally (land-only) averaged vertical profile of (a) turbulent kinetic energy (TKE) and (b) liquid cloud moisture content. MMF, SFLX2CRM, and MAML are marked in different colors using black, blue, and red, respectively.

adequately resolved. TKE in MAML is stronger than SFLX2CRM (fig.9), which explains the further reduction of condensation at the model level.

## 4   Conclusions

Here, we present a numerical study that explores 3 methods to model the local scale land-atmosphere interaction processes in the MMF version of E3SM. Traditionally in earth system models, an atmosphere and a land is coupled at each large-
scale grid, which is generally in the order of 100km. Land surfaces are characterized by land cover, terrain, and soil texture, which are naturally heterogeneous across various spatial-temporal scales. Therefore, too large coupling scale between land and atmosphere can easily undermine the importance of land-atmosphere interactions and their impact on convective cloud formations. The MMF allows a global atmospheric model to run at a cloud-resolving scale, which gives us a motivation to explore the impact of coupling land-atmosphere at cloud resolving scales on the energy and water budget at land surface.
Alongside the traditional method of coupling land-atmosphere in E3SM, two strategies are assessed at global scale from Baker et al. (2019), both using 2-D CRM with interactive land surface. Therefore, these two methods can exchange energy and water directly between CRM and a land model. First method exchanges a CRM domain averaged values with a single instance of land model, therefore only homogeneous interactions are allowed (SFLX2CRM). Second method allows an intra-CRM





heterogeneity by coupling each CRM grid with its own land model. In our study, we used 2-D CRM with 32 grid points, and
each of these 32 CRM columns has its own land surface (MAML). By analyzing the 5-year model output, we see that the
model simulation demonstrates the positive feedback between soil moisture, evaporation, PBL humidity and precipitation as
stronger precipitation is observed over the areas with higher soil moisture. However, we find that MAML tends to produce
drier and warmer surface weather. In MAML, the warmer temperature increases the transpiration of the rainforest while there
is insignificant change in the direct soil evaporation. However, the evaporation of the moisture stored on vegetation surfaces
decreases and this reduction overpowers the increase of transpiration. On the other hand, for C3, C4 grasses and crop fields tend
to be more sensitive to the soil moisture when determining the total evaporation. As MAML simulation produces lower soil
moisture, total evaporation over grasses and crop-fields also decreases. Therefore, the total evaporation is reduced regardless
of vegetation types in MAML in comparison to MMF and SFLX2CRM. The future study can do follow-up investigation on
sensitivity of precipitation to the soil moisture in E3SM-MMF.

Current model configuration of MAML framework, where each land instance is configured with the same land surface
characteristics, produces too weak heterogeneity in land-atmosphere interactions. Therefore, we do not see any drastic changes
in the precipitation and cloud formation. However, this work provides a modeling framework in which MAML can be used
an an advanced modeling tool. In this framework, it is simple to prescribe each ELM instance with different land surface
characteristics. However it is non-trivial to prescribe realistic heterogeneity in land surface characteristics for 2-D modeling
space. Therefore, additional study can help us to investigate the role of the truly heterogeneous land surface characteristics in
land-atmosphere interactions in a global scale model

*Code and data availability.* E3SM-MMF source code with MAML implementation and data used to make figures can be accessed at the
following link: https://doi.org/10.5281/zenodo.6554887

*Author contributions.* Jungmin Lee and Walter M. Hannah contributed to the heterogeneous land-atmosphere interaction modeling concep-
tion and design, and they together wrote the software code. Model simulations and data analysis were performed by Jungmin Lee. The first
draft of the manuscript was written by Jungmin Lee and both authors commented on previous versions of the manuscript. David C. Bader
was responsible for funding acquisition and oversaw project delivery

*Competing interests.* Authors declare that they have no conflict of interest

*Acknowledgements.* This research used resources of the National Energy Research Scientific Computing Center, which is a DOE Office of
Science User Facility supported under Contract No. DE-AC02-05CH11231. This research was supported by the Exascale Computing Project
(17-SC-20-SC), a collaborative effort of the U.S. Department of Energy Office of Science and the National Nuclear Security Administration.





This work was performed under the auspices of the U.S. Department of Energy by Lawrence Livermore National Laboratory under Contract DE-AC52-07NA27344





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
