# Peer review of "Representation of atmosphere induced heterogeneity in land – atmosphere interactions in E3SM-MMFv2"

_Geoscientific Model Development, 2023_

## Author Comment (AC2)

$$BFLX = \frac{SHF}{c_p} + 0.61T_s\frac{LHF}{L_v}$$

---

## Author Response (AR1)

**RC #1**

Review of GMD submission gmd-2023-55 "Representation of atmosphere induced heterogeneity in land-atmosphere interactions in E3SM-MMMFv2" by Le et al.

Recommendation: accept *after major revisions*

The manuscript reports analysis of simulations applying superparameterized GCM, the MMF, with three different couplings between atmospheric model and the land-surface model. Overall, I feel this manuscript reports analysis that is superficial and can be significantly expanded to make the comparison more meaningful. In addition – if I understand the model setup correctly – the setup does not allow a possibly significant aspect of the atmosphere-land surface interactions through the development of the subgrid-scale contrasts of surface characteristics. Finally, I am not sure if the GMD is the proper journal to report the simulations results. Below I discuss those points in more detail. The paper should be accepted only after those points are appropriately addressed. I may have more comments after the major issues are addressed and the manuscript is revised.

Authors would like to thank the reviewer for his time and effort in providing us the valuable comments and suggestions. Our respond is colored blue below.

Specific major comments:

1. I will leave it to the editor to decide if GMD is the appropriate journal for this submission. I personally feel JAMES would be more appropriate as the paper does not report any model development, just the impact of various possible couplings between the GCM's atmospheric and land-surface components.

2. The analysis focusses on temporally and spatially averaged data. However, over land, the key is the diurnal cycle and how the surface characteristics impact it through surface heat fluxes. In essence, Bowen ratio of the surface heat flux impacts the surface buoyancy flux and this has a strong impact on daytime convection development. The paper that comes to mind (and references therein) is:

Thomas, L., N. Malap, W. W. Grabowski, K. Dani, and T. V. Prabha, 2018: Convective environment in pre-monsoon and monsoon conditions over the Indian subcontinent: the impact of surface forcing. Atmos. Chem. Phys., 18, 7473-7488.

So the first suggestion is to look at the differences in model results of the diurnal cycle, not just at averaged heat fluxes between 6 to 18 local time. The analysis reported in Yang and Slingo (2001) can be repeated using model results with the focus on land.

Yang, G., and J. Slingo, 2001: The Diurnal Cycle in the Tropics, Mon. Wea. Rev., 129, 784-801.

Authors thank you for the suggestion for further analysis. We looked into the diurnal cycle of precipitation; however, we did not see significant differences in the diurnal precipitation (both phase and amplitude) regardless how we couple land and atmosphere models. Figure below shows the Evans plot of the diurnal cycle of precipitation. Hue represents diurnal maximum timing and saturation represents diurnal amplitude. As shown in the figure below, there is slight difference in the diurnal maximum timing in Amazon and central Africa, but they are not significant. However, in our daytime mean analysis, we do find that the surface buoyancy is stronger in MAML case (Figure 10), which helps to clear up the lowest model level condensation (Figure 11). This is the most significant impact that we could find using three difference land-atmosphere coupling method.

[Figure]

The second suggestion is that one should look at the differences at various climatic zones, for instance, tropical versus subtropical versus summertime extra-tropics, and versus dry and rainy season in the tropics and subtropics. Analysis along those lines would make modeling results more convincing. I appreciate the fact that such information is to some extent provided by the global maps (e.g., Fig. 2), but selecting specific areas (e.g., US Midwest, Amazon, India, Europe, etc.) and showing local maps would bring some interesting differences.

What we find by inspecting each cases (MMF, SFLX2CRM, and MAML) from Figure 1 through 6 is that the most pronounced differences are located over the tropics where the surface net radiation is high, where there is less seasonal differences. We do find that in those areas, change in soil moisture, change in EF, change in near surface specific humidity and precipitation is positively correlated. We show this by presenting the scatter plot between those variables over in Amazon as an example and we added the figure 8 (shown below) in the text. Therefore, instead of focusing on specific regions separately, our analysis focuses on the tropical belt where the difference is relatively significant in comparison to other areas.

[Figure]

**Figure 8. Scatter plots of 5-day averages of (a) near surface soil moisture versus evaporative fraction, (b) evaporative fraction versus near surface specific humidity, (c) near surface specific humidity versus precipitation averaged over Amazon. MMF, SFLX2CRM, and MAML are denoted by black, blue and red dots, respectively.**

3. As far as I understand the model setup, land surface models are not allowed to develop small-scale variability in the MAML setup. I think this is a significant drawback of the methodology not discussed in the paper. To what extent such variability is important for the problem the paper investigates? For instance, if rain falls in a fraction of the CRM domain (e.g., because of convection), then the land surface is modified beneath those CRM columns. As a result, CRM can develop circulations that can affect subsequent evolution of small-scale flow and subsequent convection development. This is not possible in the current setup, correct? That said, the wind that plays important role in atmosphere - land surface interactions seems to be taken from the GCM (per l. 71). Again, this is unrealistic.

In SFLX2CRM case, yes there is no intra CRM heterogeneity in the land surface model. Entire CRM **domain** is coupled to a single land model, therefore even if there is precipitation in one part of the domain, the land model won't feel this heterogeneity in the spatial distribution of precipitation as an input to drive land surface processes. However, in MAML, we couple each CRM **grid** to an independent copy of land model.

For instance, in our set up, each CRM domain is configured with 32 grid points. Entire CRM domain is coupled to a 32 independent copy of land models, that is, one land model per CRM grid. Therefore, if only a portion of CRM grid develops precipitation, only the land model beneath those specific CRM grid will retrieve those precipitation as an input to drive land surface processes. In MAML, atmosphere induced heterogeneity in land-atmosphere coupling inside CRM domain is represented. We add this information in section 2.2 and also in conclusion.

In the MMF, we use 2D CRM, therefore meridional wind component can be unrealistically strong. As a land model input, we need wind speed to compute surface fluxes. We were concerned that using the wind speed computed directly from the 2D CRM wind fields would yield unrealistic surface fluxes. As an alternative we used GCM wind speed near surface to pass onto the land model. We did not test the implication of using GCM wind to compute surface heat fluxes. We added this information in L129-131 and is given below:

L129-131: Using EAM wind to drive land surface processes and grouping neighboring CRM columns for radiation calculation decreases heterogeneity induced by atmosphere in MAML. We did not test how these modifications would affect the resulting climate.

There is also another issue that requires a little more detailed explanation of the coupling between GCM and CRM models. I assume CRMs are aligned in a specific way and predict small-scale fluctuations of only one horizontal wind component (i.e., along the 2D domain). But the total wind can be taken from the GCM to create local (i.e., CRM's) 2D surface winds. Such winds should be used in the land model to calculate surface fluxes.

In E3SM-MMF, there is large scale forcing from GCM to CRM but no momentum feedback. Therefore, CRM wind is effectively nudged toward GCM wind rather than coupled. Therefore, we had to use GCM winds.

4. This comment is related to 2 and 3 above. For the diurnal cycle over land, it has been shown in high-resolution simulations that the interactive surface fluxes (in contrast to the prescribed horizontally-homogeneous surface fluxes) have a noticeable impact on moist convection diurnal cycle (e.g., Kurowski et al. 2018). It is unclear to me if that aspect can be considered in the way models are coupled in the current setup. This might be important for the land-atmosphere coupling. Finally, the CRM does not include CRM-scale topography. This is an important drawback of the atmosphere-land coupling in MMF. Perhaps this should be appropriately stressed in the manuscript.

Kurowski, M. J., K. Suselj, W. W. Grabowski, and J. Teixeira, 2018: Shallow-to-deep transition of continental moist convection: cold pools, surface fluxes, and mesoscale organization. J. Atmos. Sci., 75, 4071-4090.

Yes I am aware of previous CRM and LES studies that looked at the heterogeneous surface fluxes that can induced secondary mesoscale circulations and indeed impact on the water cycle over land. However, in the current set up of MAML, even if each CRM grid is coupled to its own land surface model, land surface characteristics across the multiple land surface models within a CRM domain remains the same. Therefore, the set up is missing the heterogeneity induced from the heterogeneity in the land surface characteristics including topography. In MAML framework, where each land model instance can read in its own initial land condition file, theoretically, we can prescribe difference land surface types to each land model instance within a given CRM domain. However, we are using 2D CRM, therefore translating the observed land surface heterogeneity into the 2D space is a non trivial matter and out-of-scope of this study. We modified the existing sentence to stress that the CRM does not include CRM-scale topography in Line 127-129.

Line 127-129: These nx copies of ELM are prescribed with the same land surface characteristics. Also, CRM does not include CRM-scale topography. Therefore, this method does not fully represent a significant level of surface heterogeneity.

**RC #2**

Representing land-atmosphere interactions is crucial for weather and climate models. Lee et al., manuscript implements three different approaches to represent land-atmosphere interactions at different spatial scales within the multi-scale modeling framework (MMF) of DOE E3SM model. It also shows the difference in 5-year climatology between the simulations with these three approaches. This manuscript provides valuable guidance on the model development in sub-grid land-atmosphere coupling and helps understand the soil-moisture, PBL, and precipitation interaction mechanisms. My comments are shown below.

**Major comments**

1. The current figures should be presented in different orders. Sec. 3.1 covers Figure 2-6, and Sec. 3.2 starts with Figure 2 again. Readers have to go back and forth to look up the correct figures. Thus, the figures for climatology from MMF simulations should be grouped together and placed first, followed by the difference between MMF and SFLX2CRM, and MAML simulations.

   We appreciate your suggestion. Many times, in the text, we try to point out the regional changes are interconnected among different variables to qualitatively represent land-atmosphere coupling. Even if we group global climatology and differences together, readers should still have to go back and forth to follow the text. We apologize for this inconvenience, but we would like to leave the order as it is.

2. I agree that MAML makes minor changes in 5-year climatology simulations on a global basis as shown in Table 1. However, the spatial figures do show some non-negligible regional changes. For instance, the changes in CRES and precipitation over Amazon, North America, East Asia, Central Africa and etc. in Figure 6 (e) and (f) show noticeable differences (up to ~20% estimated from the naked eye). Therefore, in addition to the global mean change, the authors should emphasize the regional changes in the text, at least in the Abstract and Conclusion parts.

   We thank you for the suggestion. We added a couple sentences throughout the text to highlight the regional differences. Those changes are given below

L12-13: 5-year global simulations are run using these 3 coupling strategies and we find some regional differences but overall small change whether a land model is coupled to CRM or a global atmosphere.

L191-192: For example, precipitation and cloud radiative effect at surface shows noticeable differences in SFLXCRM and MAML over Amazon, North America, East Asia, Central Africa regions.

We also provide additional regional differences based on the vegetation types throughout the text. Since the largest difference between model configurations exists in the tropics where the net surface radiation is the largest, we do focus on those region as a whole

3. The authors attribute the differences between the MAML and MMF simulations only to the change in heterogeneity in land-atmosphere coupling over the CRM domain. This is not the whole story. The difference is also due to the different ways of coupling between MMF and SFLX2CRM. And based on Figures 2-6, the differences between MMF and SFLX2CRM (MMF-SFLX2CRM) are larger than that due to the change in heterogeneity in land-atmosphere coupling over the CRM domain (MAML - SFLX2CRM). So the authors should discuss more on the MMF-SFLX2CRM difference. What are their differences in terms of the land-atmosphere coupling in the model? How does that translate to the difference in climatology as shown in Figure 2-6? The current manuscript fails to give a clear answer.

Global mean maps shows some areas in SFLX2CRM with larger difference than the areas in MAML, but the global means of SFLX2CRM is very similar to the means of MMF as shown in Table 1. Also, We discuss the SFLX2CRM-MMF difference focusing on the buoyancy, TKE and cloud liquid water difference in Section 3.4. In section 3.4, we discuss that the buoyancy and TKE changes with interactive land surfaces, in our case both SFLX2CRM and MAML. But the atmosphere induced heterogeneity in MAML drives stronger PBL turbulence. We also modified a sentence in L300-301 in a text to stress our point.

L300-301: We find that global means of SFLX2CRM is similar to that of MMF. However, MAML tends to produce drier and warmer surface weather.

**Minor comments**

Line 71. and Line 76-77. The authors state "use the wind velocities from EAM in all three methods" and "grouped 2 neighboring CRM columns". Does this mean that the

heterogeneity of wind and radiation is reduced in the model? How would that affect the conclusions in the manuscript?

Yes, heterogeneity of wind and radiation was reduced in the MAML due to the fact that EAM wind was used to force the land model and we grouped 2 neighboring CRM columns for radiation computation. We decided to use EAM wind because the possibility of too strong meridional wind component of 2-D CRM. We had no other option but to use EAM wind. Also, grouping 4 neighboring CRM columns for radiation was tested against no grouping case, we found negligible differences in the result climate. We didn't test how these choices would affect the land-atmospheric interaction processes in the model. We added below sentences in the text to include that information.

L129-131: Using EAM wind to drive land surface processes and grouping neighboring CRM columns for radiation calculation decreases heterogeneity induced by atmosphere in MAML. We did not test how these modifications would affect the resulting climate.

Line 95-97. Not clear what the differences are between MMF and SFLX2CRM in terms of implementation. The standard MMF method does not allow the development of surface buoyancy in the PBL?

In the standard E3SM-MMF, buoyancy is not directly passed to the CRM. Instead, buoyancy is passed only to EAM. Temperature/moisture profiles in EAM changes according to the given buoyancy. Then CRM feels the buoyancy indirectly by the large scale forcing from EAM. On the other hand, in SFLX2CRM, we pass buoyancy directly to CRM as a bottom boundary condition. So in SFLX2CRM, PBL development is directly driven by the surface buoyancy. This information is found in L111-112 and L118-120.

Line 128. Not exactly. The strength of sensible heating is also related to solar radiation. For instance, the sensible heating flux is low over the Antarctic with no vegetation cover.

I meant to say the baresoil area in the tropical, sub-tropical regions such as deserts produces stronger sensible heat flux. I modified the sentence as given below:

L159-160: On the other hand, regions in the tropics and the sub-tropics with no vegetation cover (1) produce stronger sensible heating as shown in fig.2a.

Line 144. Change "signification" to "significant".

Thank you for catching this typo. We changed 'signification' to 'significant' in Line 170 in the revised manuscript.

Table 1. The last column should be "MAML"?

Thank you for catching this typo. We changed 'MMF' to 'MAML' in the last column of Table 1.

Line 160. "Fig. 3" is meant to be "Fig.2"?

Thank you for catching this typo. We changed 'Fig 3' to 'Fig 2' in Line 187 in the revised manuscript.

Line 172. How did the authors get the number of 3.8 W/m$^2$?

It's computed based on the average direct vegetation evaporation on broadleaf evergreen tropical trees. It's the difference between MMF and MAML case.

Figure 7. Can the authors add legends to show what the scatter dots with different colors stand for? As well as to clarify what different pft numbers are corresponding to?

Thanks for the suggestion. We added color dot legends in the figure panel, and added vegetation type instead of pft in the figure.

Line 187. It is not straightforward to see "how the lower-atmosphere condition is influenced by different coupling methods" from Figure 5. Can the authors elaborate on that?

We edited the sentence as given below:

L230-232: Fig. 5 shows how the lower-atmosphere meteorological condition is influenced by underneath surface heat fluxes and soil moisture. Regions with higher (lower) evaporation aligns with regions with humid (dry) PBL.

Line 188-192. These are qualitative descriptions of the soil moisture-PBL-precipitation interactions. Can the authors quantify the interactions to show how different coupling strategies affect these interactions?

To show the correlation between each segment of soil moisture – PBL – precipitation interactions, we added a new figure shown below. This scatter plot shows that there is positive correlation exists between EF and SOILWATER_10CM, SOILWATER_10CM and

QBOT, and QBOT and Precipitation. We added the figure shown below into the manuscript and added additional sentences in Line 233-239, which is also given below.

[Figure]

L233-239: Figure 8 shows scatter plot for each segment of soil moisture-PBL-precipitation interactions. These scatter plot presents that positive correlation exists between soil moisture and evaporative fraction (EF), EF and near surface humidity, and near surface humidity and precipitation. There is no noticeable differences between each cases.

Line 197-198. "strong spatial heterogeneity in land surface process is found where the MAML differs from the MMF the most". Can the authors explain more on how did they reach this conclusion?

We modified the sentence, and the new sentence is given below:

We use the standard deviation as a measure of CRM scale spatial heterogeneity (Fig. 8). Strong spatial heterogeneity in land surface processes is found where the MAML differs from the MMF the most by visually inspecting Fig. 2, 4-6.

Line 202. The standard deviation alone cannot reflect the difference well in spatial heterogeneity between different variables in the lower atmosphere and the land states. Instead, the standard deviation normalized by the land grid mean is a better way for the comparison between different variables.

Yes, In some cases, we agree that normalized standard deviation may be a better way for the comparison between different variables. But here, sensible heat flux, latent heat flux, and precipitation has regions where the land grid mean is close to zero, thus, gives us very strong normalized deviation regardless that the non-normalized standard deviation is very small or not. We could see large normalized standard deviation because the land grid mean value is very small, which is against what we are trying to show. Therefore, we decided to leave the figures as is. Thank you for your suggestion.

Line 208. How is the buoyancy flux diagnosed?

Buoyancy flux is diagnosed inside the code as

$$BFLX = \frac{SHF}{c_p} + 0.61 T_s \frac{LHF}{L_v}$$

Where SHF (LHF) is sensible (latent) heat flux, $c_p$ is specific heat of air, $T_s$ is near surface temperature, and $L_v$ is latent heat of vaporization. For MAML case, BFLX is computed for each land model instance. We added above information in L259-263.

Figure 10. Why do the MMF and SFLX2CRM simulations show almost the same TKE profiles, but quite different liquid cloud contents?

It is the difference between whether surface buoyancy flux is supplied as a bottom boundary condition to stir turbulences in PBL or not. In MMF, CRM does not receive the surface heat flux information in the code, therefore computes TKE inside CRM based on the vertical gradient of thermodynamic profiles. Whereas in SFLX2CRM, surface heat flux is directly coupled to CRM so surface heat fluxes are used to compute the TKE at the bottom most interface level. You can see that TKE profile is slightly higher in SFLX2CRM near the bottom. The reason why TKE profile is similar between MMF and SFLXCRM above the surface level can be explained that TKE is computed based on the thermodynamic profile and both cases have similar thermodynamic profiles.

We added below information in the text:
L282-285: In MMF, CRM does not receive surface heat fluxes from ELM, therefore computes TKE based on the thermodynamic profile. Since MMF and SFLX2CRM has similar thermodynamic profile, both produces similar TKE profile. However, near the surface, SFLX2CRM has slightly higher TKE due to the non-zero surface buoyancy flux. This is the reason why there is a large difference in the liquid cloud water profile when the TKE is very similar.

---

## Referee Report (RR1)

Review of revised GMD submission gmd-2023-55 "Representation of atmosphere induced heterogeneity in land-atmosphere interactions in E3SM-MMMFv2" by Le et al.

Overall, I am disappointed with the revision. The revised manuscript shows some minimal changes when compared to the initial submission. Maybe this is because there is not that much to show in terms of the difference between various model configurations. However, I do not think so.

Please think about the bigger picture: The key conclusion of this investigation can be summarize as follows: coupling copies of the land model to all columns of the embedded small-scale models (i.e., many land models per a GCM column) rather than using GCM fields to drive just one land model in a GCM column gives very little difference when averaged over long time (years). Does that imply that small-scale land-atmosphere coupling is irrelevant for climate? Or maybe it shows limitations of the superparameterization approach? Or maybe some differences (that I expect are there) smooth out when averaged over time? My vote is "no" for the first question, and "yes" for the question two and three. The simulations discussed in the paper should be capable to provide answers to those questions as well.

Right now, the paper shows that the results averaged over 5 years show very little difference (e.g., table1). However, I suggested in my first review that the authors looked at shorter-time scale processes, such as diurnal cycle or the impact of interactive surface fluxes on convection development over tropical or warm-season midlatitude continents. In their responses, the authors dismiss my suggestions. They state that there is little difference in the diurnal cycle of precipitation. However, is there any impact of the small-scale land-surface model on convective development? For instance, surface characteristics should show small-scale differences (e.g., availability of the surface water) when the land model is coupled to CRM columns, correct? That should affect convection development over the next day, correct? Perhaps such differences smooth out when long-term statistics is gathered, but this remains to be shown. Moreover, the authors' response to my suggestion to look at the role of interactive surface fluxes seems awkward. I do not suggest to use a different land-surface type is each CRM column. However, a small-scale precipitation pattern should develop gradients of the surface characteristics (soil moisture in particular) even if the same land surface type is used across all CRM model columns.

In summary, I still maintain that the analysis presented in the paper is superficial. One way to make it more interesting would be to contrast short time scale processes (as briefly discussed above) and long-time averages. In addition, looking at global maps is rather uninteresting way to point out differences. Can the authors be a little more creative? For instance, select various land-surface types in a given climatic zone, and select some characteristics for each land-surface type. Something along Fig. 7, for instance.

Despite my criticism, I do not want to delay publication of this manuscript. So my recommendation is to accept after minor revisions. I have several specific minor comments that the authors should address before the manuscript is accepted. Many of those comments apply to the initial submission as well. I did not report them as I thought the major issues needed to be addressed first.

Specific comments:

1. This comment was addressed by neither the authors nor the editor: "I will leave it to the editor to decide if GMD is the appropriate journal for this submission. I personally feel JAMES would be more appropriate as the paper does not report any model development, just the impact of various possible couplings between the GCM's atmospheric and land-surface components."

I have not received any correspondence from the editor either. So I assume the paper is still assumed to be considered as a GMD submission. I maintain the submission is more appropriate for JAMES.

2. There are numerous small editorial problems. Some of them were in the original submission, perhaps some are new. Below is a list (Lx means the comment applies to line x in the manuscript):

2a. The land surface model is not described/discussed. Please add.

2b. Throughout the text: I am not sure the term "instance" is the best way to describe application of the land model. Maybe "copy" would be better?

2c. L9: "…coupling multiple land instances to each column..". I think this incorrect. Only one copy of a land model (or scheme) is applied to each CRM column, correct?

2d. L19: "Careful" is not a good word here. Please remove as it is not needed.

2d. L28: "processes … controls"?

2e. L35: "…to 25 generalize"?

2f. L37: "landatmosphere"?

2g. L74: EAM is not defined.

2h. L80: "resolution" should be "grid length" or "grid spacing".

2i. L81: "number of vertical model level is"?

2j. L84 and L87: "centered year 2000"? "To avoid this caveat"?

2k. L89: "ELM" is not defined.

2l. L115/116: The sentence "This method…". First, I think all coupling methods prescribe surface buoyance flux, either indirectly (like the MMF method) or directly, like the two other. In the first two methods, the surface buoyancy forcing is horizontally uniform. In the third method, it can be horizontally heterogeneous. If this is incorrect, then there is something in the methodology that I do not understand. Also "turbulences" is not a word.

2m. L.123: Replace "…are prescribed with…" with "feature". Also "nx" is not defined.

2n. L. 140: "period"?

2o. Daytime is defined in the manuscript as the average between 6 and 18 local hour. This is not appropriate for wintertime extratropics where the daytime is much shorter. Perhaps making averages over periods with positive incoming solar radiation would make more sense. If this is too much trouble, just commenting on that would be sufficient.

2p. L. 146/147: I dot understand what is meant by "…magnitudes of fluxes increase…". The maximum increase? The range (night time versus daytime) increase? Please explain.

2q. L173: "…terrain effect and local-scale land-atmosphere interaction process…". Explain what you mean by that statement. What is "terrain effect"? Are "local-scale land-atmosphere interactions" insignificant in other geographical locations?

3. Since the paper use several acronyms (some not defined as pointed out above), I suggest to include an acronym table to help the reader.

4. Calculation of the surface fluxes in CRM, lines 85 to 87. I do not understand this logic. To me, the correct way to couple GCM and CRM winds for the surface flux calculation is to combine horizontal wind from the 2D CRM with the second GCM wind component. For instance, is the CRM is aligned along the zonal direction, the wind used in the surface flux formula at each CRM column should be taken as sqrt[(u(x)**2 + V**2)], where u is the surface horizontal wind in the CRM, and V is the meridional wind from the GCM model at the location of the embedded CRM. If this is not correct, then please explain how this is done in the code and why it is not done in the way I explain above.

5. Fig. 10. I do not see any black point in the figure. Maybe use three panels or artificially separate clouds of points. I think the difference between red points and all others attest to the role of local circulations that develop because of the horizontal variability of surface characteristics in MAML approach, correct?

6. Fig. 11. Left panel: would it make more sense to average TKE only over locations with strong surface buoyancy flux (for convective situations) or strong surface shear of the mean wind (for shear-driven boundary layer)? The plot shows results over just a few levels in the BL and I am not sure what to think about the significance of that plot. Right panel: I think it shows that circulations that develop in the MAML setup help to remove the "stratofogulus" (e.g., https://doi.org/10.1029/2020JD032619) that often develops in climate model simulations when boundary layer circulations are inefficient in transporting water vapor from near the surface to higher levels.

== END ==

---

## Author Response (AR2)

Review of revised GMD submission gmd-2023-55 "Representation of atmosphere induced heterogeneity in land-atmosphere interactions in E3SM-MMMFv2" by Le et al.

Overall, I am disappointed with the revision. The revised manuscript shows some minimal changes when compared to the initial submission. Maybe this is because there is not that much to show in terms of the difference between various model configurations. However, I do not think so.

Please think about the bigger picture: The key conclusion of this investigation can be summarize as follows: coupling copies of the land model to all columns of the embedded small-scale models (i.e., many land models per a GCM column) rather than using GCM fields to drive just one land model in a GCM column gives very little difference when averaged over long time (years). Does that imply that small-scale land-atmosphere coupling is irrelevant for climate? Or maybe it shows limitations of the superparameterization approach? Or maybe some differences (that I expect are there) smooth out when averaged over time? My vote is "no" for the first question, and "yes" for the question two and three. The simulations discussed in the paper should be capable to provide answers to those questions as well.

Right now, the paper shows that the results averaged over 5 years show very little difference (e.g., table1). However, I suggested in my first review that the authors looked at shorter-time scale processes, such as diurnal cycle or the impact of interactive surface fluxes on convection development over tropical or warm-season midlatitude continents. In their responses, the authors dismiss my suggestions. They state that there is little difference in the diurnal cycle of precipitation. However, is there any impact of the small-scale land-surface model on convective development? For instance, surface characteristics should show smallscale differences (e.g., availability of the surface water) when the land model is coupled to CRM columns, correct? That should affect convection development over the next day, correct? Perhaps such differences smooth out when long-term statistics is gathered, but this remains to be shown. Moreover, the authors' response to my suggestion to look at the role of interactive surface fluxes seems awkward. I do not suggest to use a different land-surface type is each CRM column. However, a small-scale precipitation pattern should develop gradients of the surface characteristics (soil moisture in particular) even if the same land surface type is used across all CRM model columns.

In summary, I still maintain that the analysis presented in the paper is superficial. One way to make it more interesting would be to contrast short time scale processes (as briefly discussed above) and long-time averages. In addition, looking at global maps is rather uninteresting way to point out differences. Can the authors be a little more creative? For instance, select various land-surface types in a given climatic zone, and select some characteristics for each land-surface type. Something along Fig. 7, for instance.

Despite my criticism, I do not want to delay publication of this manuscript. So my recommendation is to accept after minor revisions. I have several specific minor comments that the authors should address before the manuscript is accepted. Many of those comments apply to the initial submission as well. I did not report them as I thought the major issues needed to be addressed first.

Specific comments:

We appreciate the comments and suggestions given by the reviewer. Please see our response in blue below.

1. This comment was addressed by neither the authors nor the editor: "I will leave it to the editor to decide if GMD is the appropriate journal for this submission. I personally feel JAMES would be more appropriate as the paper does not report any model development, just the impact of various possible couplings between the GCM's atmospheric and land-surface components."

I have not received any correspondence from the editor either. So I assume the paper is still assumed to be considered as a GMD submission. I maintain the submission is more appropriate for JAMES.

We appreciate the reviewer's suggestion that this manuscript would be more suitable for JAMES. However, this paper was intended to introduce 3 different methods in land-atmosphere coupling within E3SM-MMF and provide initial comparison of the impact of those methods on long-term climate. Our result suggests that the way land-atmosphere is coupled in E3SM-MMF leads to rather insignificant impact on long-term climate even though significant more computation resources are required to enable interactive land-atmosphere interaction at CRM grid level. Therefore, we find GMD is an appropriate journal to publish our manuscript.

2. There are numerous small editorial problems. Some of them were in the original submission, perhaps some are new. Below is a list (Lx means the comment applies to line x in the manuscript):

2a. The land surface model is not described/discussed. Please add.
Thank you for the suggestion. We added information about the land surface model (ELM) in line 82-89 and is given as below:
L82-89: E3SM Land Model (ELM) inherits many of its functionalities from its source model, the Community Land Model version 4.5 (CLM4.5; Oleson et al., 2013). ELM simulates hydrological and thermal operations in vegetation, snow, and soil for a variety of land cover types including bare soils, vegetated surfaces, lakes, glaciers, and urban areas. Leaf area index is determined utilizing satellite data and photosynthesis without any constraints from leaf nutrients. Since branching off from CLM4.5, ELM has undergone various improvements (Golaz et al. 2019). The impact of aerosol and black carbon on snow was added. The evaporation was reduced over pervious road under dry condition. The equation for stomatal conductance was revised to avoid inaccurate representation of negative internal leaf $CO_2$ concentrations. Also, the nighttime albedo over land was updated to 1.

2b. Throughout the text: I am not sure the term "instance" is the best way to describe application of the land model. Maybe "copy" would be better?

For MAML, we modified the ensemble simulation capabilities of E3SM. From there, we inherited the term "instances". However, I see that "copy" would be more suitable. So, I converted all 'instances' to 'copies' in the text.

2c. L9: "…coupling multiple land instances to each column..". I think this incorrect. Only one copy of a land model (or scheme) is applied to each CRM column, correct?
Thank you for catching this. We revised the sentence in line 9 and is given below:
3) coupling a single copy of land model to each column of the CRM grid (MAML).

2d. L19: "Careful" is not a good word here. Please remove as it is not needed.
Thank you for the suggestion, We removed it in L19

2d. L28: "processes … controls"?
Thank you for catching the typo. We changes "controls" to "control" in L28

2e. L35: "…to 25 generalize"?
We removed '25' in L37

2f. L37: "landatmosphere"?
In the word document version that I have has '-' between land and atmosphere. This could be an error occurred during conversion of docx into pdf.

2g. L74: EAM is not defined.
Thanks for catching it. We defined EAM, and is given below as:
E3SM Atmospheric Model (EAM) in L80.

2h. L80: "resolution" should be "grid length" or "grid spacing".
Thank you for the suggestion, we replaced 'resolution' with 'grid spacing' in L93.

2i. L81: "number of vertical model level is"?
Thank you for catching the typo. We changed 'level' in the sentence to 'levels' in L94.

2j. L84 and L87: "centered year 2000"? "To avoid this caveat"?
We changed 'centered year 2000' to 'centered at year 2000' in L96, also changed 'to avoid this caveat' to 'to bypass this difficulty' in L100.

2k. L89: "ELM" is not defined.
ELM is defined as 'E3SM Land model' in L82.

2l. L115/116: The sentence "This method…". First, I think all coupling methods prescribe surface buoyance flux, either indirectly (like the MMF method) or directly, like the two other. In the first two methods, the surface buoyancy forcing is horizontally uniform. In the third method, it can be horizontally heterogeneous. If this is incorrect, then there is something in the methodology that I do not understand. Also "turbulences" is not a word.
Your understanding of the method section is correct. We modified the sentence in L132-133 and is given below as:
This method prescribes surface buoyancy forcing that is horizontally homogeneous.

2m. L.123: Replace "…are prescribed with…" with "feature". Also "nx" is not defined.

Thanks for the suggestion. We replaced 'are prescribed with' with 'feature' in L144. We also added definition of nx – number of horizontal grids – in line 137.

2n. L. 140: "period"?

Thank you for catching the typo. We changed 'periods' to 'period' in L161.

2o. Daytime is defined in the manuscript as the average between 6 and 18 local hour. This is not appropriate for wintertime extratropics where the daytime is much shorter. Perhaps making averages over periods with positive incoming solar radiation would make more sense. If this is too much trouble, just commenting on that would be sufficient.

 Thanks for the suggestion. We commented about this in Line 168-169 and is given below: However, one should note that the day length in extra-tropics is shorter in winter time.

2p. L. 146/147: I dot understand what is meant by "…magnitudes of fluxes increase…". The maximum increase? The range (night time versus daytime) increase? Please explain.

It means daytime mean fluxes have stronger magnitude than the annual mean fluxes. We rewrote the sentence in L167-168 and is given below:

L167-168: In comparison to the 5-year climatology, the magnitudes of daytime mean fluxes are higher than the amount of  annual mean fluxes,

2q. L173: "…terrain effect and local-scale land-atmosphere interaction process…". Explain what you mean by that statement. What is "terrain effect"? Are "local-scale land-atmosphere interactions" insignificant in other geographical locations?

I meant the large-scale circulation from terrain by 'terrain effect'. I revised the sentence to reflect that in L200-201. Local-scale land-atmosphere interactions are present in other geographical locations but can sometimes be muted by large-scale land-atmosphere interaction. I was emphasizing that both large-scale and local-scale land-atmosphere interaction is important factor in moisture convection in Amazon.

3.      Since the paper use several acronyms (some not defined as pointed out above), I suggest to include an acronym table to help the reader.

We added a Table A1 in Appendix section and is given below:

Line 74: This paper uses several acronyms and Table A1 is added in Appendix to help readers.

**Table A1. Table of acronyms**

| Acronym | Meaning |
| --- | --- |
| BFLX | Surface buoyancy flux |
| CRES | Net cloud radiative effect at surface |
| CRM | Cloud resolving model |
| EAM | E3SM Atmosphere Model |
| EF | Evaporative Fraction |
| ELM | E3SM Land Model |

| | |
|---|---|
| E3SM | Energy Exascale Earth System Model |
| GCM | Global climate model |
| LHFLX | Latent heat flux |
| MMF | Multi-scale modeling framework |
| PBL | Planetary boundary layer |
| PFT | Plant functional type |
| Q2M | Specific humidity at 2-meter height |
| RH | Relative humidity |
| SAM | System for atmospheric modeling |
| SHFLX | Sensible heat flux |
| SST | Sea-surface temperature |
| TKE | Turbulence kinetic energy |
| TSA | Temperature at 2-meter height |
| TSOI_10CM | Soil temperature in the upper 10 cm |

4.      Calculation of the surface fluxes in CRM, lines 85 to 87. I do not understand this logic. To me, the correct way to couple GCM and CRM winds for the surface flux calculation is to combine horizontal wind from the 2D CRM with the second GCM wind component. For instance, is the CRM is aligned along the zonal direction, the wind used in the surface flux formula at each CRM column should be taken as sqrt[(u(x)**2 + V**2)], where u is the surface horizontal wind in the CRM, and V is the meridional wind from the GCM model at the location of the embedded CRM. If this is not correct, then please explain how this is done in the code and why it is not done in the way I explain above.

Unfortunately, we can't combine wind components from GCM and CRM to get the wind speed for the land model. It is because doing so violates the conservation of mass. Also, the friction is applied to GCM winds, so we need the input winds to the surface components to match.

5.      Fig. 10. I do not see any black point in the figure. Maybe use three panels or artificially separate clouds of points. I think the difference between red points and all others attest to the role of local circulations that develop because of the horizontal variability of surface characteristics in MAML approach, correct?

As you understood, the purpose of this figure is to show that the red dots (MAML case) is different from the other 2 cases (SFLX2CRM, and default). In Figure 10, the distribution of black points is very similar to that of blue points, that, unfortunately, made the distinction between black and blues points hard. However, we added the regression lines with linear

regression equations on the figure and we think that serves the purpose to compare MAML (red dots) with two other cases, which are very similar. So, we will leave the figure as it is, but we appreciate your suggestion.

6.    Fig. 11. Left panel: would it make more sense to average TKE only over locations with strong surface buoyancy flux (for convective situations) or strong surface shear of the mean wind (for shear-driven boundary layer)? The plot shows results over just a few levels in the BL and I am not sure what to think about the significance of that plot. Right panel: I think it shows that circulations that develop in the MAML setup help to remove the "stratofogulus" (e.g., https://doi.org/10.1029/2020JD032619) that often develops in climate model simulations when boundary layer circulations are inefficient in transporting water vapor from near the surface to higher levels.

I wanted to show the lowest level because it shows the largest difference between MAML and other two cases. The meaning of this left panel is that MAML has stronger TKE near surface, therefore more efficient in transporting water vapor from near the surface to higher levels. For the same reason, we think sampling TKE per surface buoyancy or near surface shear is unnecessary.

We added a sentence to denote the presence of 'stratofogulus' in E3SM-MMF and is given below:

L292-294: Gettelman et al. (2020) reported that CAM5 also develops clouds in the lowest-model level layer ("stratofogulus") because boundary layer circulation is inefficient in transporting water vapor from near surface to higher levels.

[Figure]

== END ==